# Algorithm for Designing Professional Retraining Programs Based on a Competency Approach

**Rushana R. Anamova [1], Leonid V. Bykov [2,\*] and Dmitri A. Kozorez [3]**

1   Department of Educational and Methodological Support of Continuing Professional Education,
    Moscow Aviation Institute (National Research University), 125993 Moscow, Russia; anamovarr@mai.ru
2   Department of Continuing Professional Education, Moscow Aviation Institute (National Research
    University), 125993 Moscow, Russia
3   Prorector of Education, Moscow Aviation Institute (National Research University), 125993 Moscow, Russia;
    kozorezda@mai.ru
\*   Correspondence: bykovlv@mai.ru

**Abstract:** A new methodology is proposed for designing professional retraining programs for aviation, rocket, and space industry employees, focused on the formation of the necessary competencies. The novelty of the proposed method is in the formalization of the design process and the use of digital technologies. The advantage is the use of a modular principle of program design, which provides the opportunity to implement individual training paths for workers in industrial enterprises. The methodology was successfully tested at the Moscow Aviation Institute (National Research University) in the provision of educational services to enterprises in the aviation and aerospace industries.

**Keywords:** vocational education; competence-based approach; professional retraining; aerospace engineering; individual learning trajectory; digitalization

## 1. Introduction

The modern dynamics of the aerospace industry's technological base development involves the rapid retraining of the personnel of enterprises in accordance with the emergence of new design and production technologies [1,2]. Enterprises call for cooperation with educational institutions, where academic staff has the requisite knowledge and professional competencies in the field of instructional organization and support [3,4]. Higher educational institutions and, in particular, aerospace universities must answer this call [5,6].

This article presents a competence-based approach to the design and implementation of professional retraining programs (PRP) for engineering and technical personnel of enterprises used in the Moscow Aviation Institute (National Research University)—MAI (NRU) [7,8]. The goal of the examples of PRPs considered in this paper is the formation of necessary knowledge and skills in the field of managerial and production technologies of aircraft and rocket science.

The PRP development and implementation features were considered in more detail in this paper. In additional vocational education, the consumer is the customer. There are two possible options: The customer as a natural person involved in self-education and personal development alongside professional growth, and the customer as a juridical person or organization whose goal is to train employees in order to develop the industry's human resource sector [9].

In this article, the second option was considered, since fulfilling the requirements of the customer organization is a more complex and responsible task. When the organization acts as a customer, the quality of the education results is determined by the level of conformity of the competencies formed

by the employee, of their knowledge and skills after undergoing training, and the expectations of the employer. The relevance of the research conducted in the article is due to the fact that the development of the content of training programs for PRPs is a task that requires an integrated approach to design, taking into account the level of training of people entering training. As a rule, PRP trainees are people who have experience in performing certain work duties, but do not have the corresponding specialized training. The task of the educational institution acting as an executor is to find the trainees' knowledge and skill gaps, eliminate them, and form special competencies necessary for the employee to perform certain labor functions specific to a particular production [10].

One of the main criteria for the success of vocational retraining of employees is to fully meet enterprise requirements from the results of the provided education services [11,12]. At the same time, the educational institution conducting the training should take into account two essential aspects: Firstly, compliance of the goals and outcomes of the training with the professional industry standards and, secondly, with the needs of certain production. Focusing not only on the requirements of professional industry standards but also on the needs of production makes it possible to develop an accurate training plan and to form the necessary competencies for trainees at the stage of developing an additional PRP [13]. Thus, the required quality of training is achieved [14,15].

The authors propose applying the conceptual design of additional professional programs to the competency-based approach used in Russian universities to study basic educational programs. This approach includes the shift of the focus away from the teacher and the content of the course ("teacher-centered approach") to the trainee and the expected results of the training ("trainee-centered approach"). This allows the trainee to develop the skills and abilities required in their professional activity.

Accordingly, it was hypothesized that the design of the PRP using the competence-based approach with a modular construction principle would improve the quality of professional retraining of employees of enterprises. The subject of this research is the methodology for the development of the PRP content. The object of the research is the professional retraining of employees.

The purpose of the study was to create a methodology for the development of the PRP content that would allow the formation of new competencies among employees of enterprises that meet the requirements of modern design and production technologies.

## 2. Methodology

The development of the content of the PRP starts with setting out the training objectives (competencies and learning outcomes).

As a rule, competency is the set of employee characteristics and skills that enable and improve the efficiency or performance of a certain type of activity [16,17]. When speaking of additional vocational education, we will define the concept of competence as the ability (or willingness) of the trainee to perform certain work functions (tasks) upon completion of the training.

The basic requirements for formulating the content of competence are the following:

1. It should be connected with the specific objectives of the PRP;
2. It should be brief and clear, with no necessity for clarification ("transparent");
3. The wording of the competence should be unambiguous;
4. If it is necessary for the listener to perform an expanded list of labor functions, the wording of the competence should combine these actions in meaning or be based on the most significant of them.

For illustrative purposes, it is worth giving several examples of statements of the competencies for two professional retraining programs implemented at the MAI (NRU): "Design, Production, Testing, and Operation of Helicopters" of the "Helicopter Design" Department and "Project Management" of the "Innovative Economics, Finance, and Project Management" Department (Table 1).

**Table 1.** Examples of formulations of competencies of courses.

| № | Name of the Course | Competencies |
|---|---|---|
| **PRP "Design, Production, Testing, and Operation of Helicopters"**, program manager: B.L. Artamonov | | |
| 1 | Helicopter fundamentals and principles of helicopter flight | Ability to apply knowledge about the vehicle and the purpose of its units, systems, and equipment |
| 2 | Aerodynamics and dynamics of helicopter flight | Ability to apply knowledge of the physical processes underlying the operation of the rotor, flight of a vertical take-off and landing of the rotary aircraft, and their mathematical descriptions |
| 3 | Helicopter production technology | Willingness to participate in the development, customization, and exploitation of technological processes during the preparation for the production of helicopter components and assemblies |
| 4 | Operational manufacturability and reliability | Ability to analyze existing systems of technical operation and offer design solutions that increase the operational manufacturability and reliability of the helicopter and its systems |
| **PRP "Project Management"**, program manager: T.I. Zueva | | |
| 5 | Cost management | Ability to make decisions on estimating project cost and project cost management at different stages of its life cycle |
| 6 | Project team management | Ability to form a project team and to manage social relations and project human resources |
| 7 | Project quality management | Ability to develop and implement quality assurance programs in project management |
| 8 | Logistic support of projects | Ability to plan organizational activities and allocate participants' functions in the field of logistics in the implementation of projects |
| 9 | Project risk management | Ability to analyze and evaluate the risks of innovative projects and to develop cost-effective measures to prevent them |

The requirements for the formulation of a competence provide the specification of the context and mechanism for the implementation of the actions indicated in the competence.

In the given example, specification was used in the formulation of competence № 2 ("What physical processes?"—"… underlying the operation of the rotor and flight of a vertical take-off and landing of the rotary aircraft") and № 4 ("What design solutions can be proposed?"—"… that increase the operational manufacturability and reliability of the helicopter and its systems").

The visual implementation of the requirements for the formulation of competence from № 4 is shown in the example of the PRP course № 6, "Project Team Management": The competence formulation lists the main actions, which are referred to as the "Team Management" function ("to form a team", "to manage social relations in a team", "to manage project human resources").

The list of competencies agreed upon with the customer that should be formed among students as a result of the training necessary for them to perform certain labor functions is, in fact, an established competency model of the student based on the learning outcomes (Figure 1).

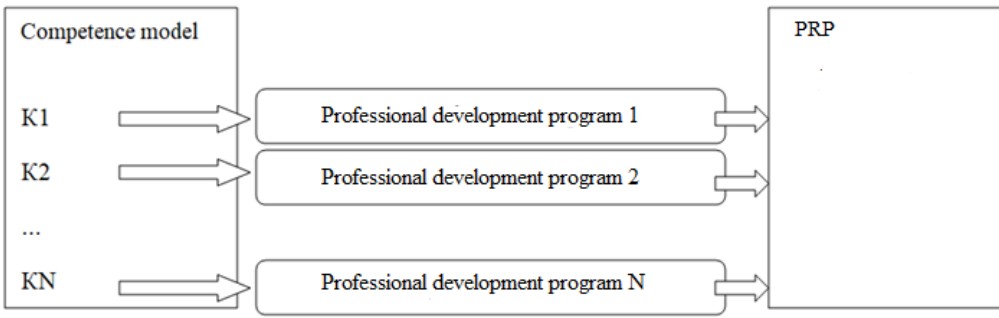

**Figure 1.** Development of the professional retraining program (PRP) content.

When formulating competencies, the components of the learning objectives (according to the taxonomy of B. Bloom [18,19] and his supporters [20]) are taken into account in three domains:

- Cognitive;
- Affective (emotion-based);
- Psychomotor.

B. Bloom proposed breaking the cognitive domain into the following six levels of objectives: Knowledge, comprehension, application, analysis, synthesis, and evaluation [21,22].

To perform the work duties given in the set of created competencies, training results must be obtained, i.e., the student must acquire certain knowledge and skills in the learning process. Training outcomes, as well as competencies, should be planned by the trainer of the program during the process of its development upon agreement with the customer.

In the implementation of PRP courses, the training outcomes are formulated depending on their belonging to the following groups:

1.  Knowledge and comprehension ("to know", "to have an idea", "to understand");
2.  Intellectual skills ("to be knowledgeable in ...", "to be able to analyze ...", "to be able to evaluate ...");
3.  Practical skills ("to have skills", "to be able", "to be able to apply knowledge in the field ...", "to have skills in the field ...").

In a similar way, we can approach the formulation of the PRP training outcomes discussed above (Table 2). The training outcomes given in the table contain formulations from the three groups mentioned above. In addition, the majority of the training outcomes constitute the ones presented in the third group ("practical skills"). This is due to the fact that during the realization of the PRP, the focus is mainly on the obtaining of the necessary practical skills in order to ensure the integration of the employee being trained into their new work environment as quickly as possible.

**Table 2.** Formulations of the training outcomes.

| № | Name of the Course | Training Outcomes |
|---|---|---|
| \multicolumn | **PRP "Design, Production, Testing, and Operation of Helicopters"**, program manager: B.L. Artamonov | |
| 1 | Helicopter fundamentals and principles of helicopter flight | 1. Knowledge of the basic principles of helicopter flight; <br> 2. Knowledge of the history of the development of rotorcraft designs; <br> 3. Knowledge of the helicopter fundamentals and systems; <br> 4. Ability to evaluate the main advantages and disadvantages of rotary-wing aircraft of various schemes; <br> 5. Knowledge of special terminology in the field of helicopter engineering. |
| 2 | Aerodynamics and dynamics of helicopter flight | 1. Knowledge of the general aerodynamics of subsonic aircraft; <br> 2. Ability to use the basics, laws, and methods of natural sciences to solve the problems of aerodynamic calculation and flight dynamics of rotary-wing aircraft; <br> 3. Knowledge of the basics of aerodynamics and flight dynamics of single-rotor and coaxial helicopters; <br> 4. Ability to apply approximate methods and algorithms for aerodynamic calculation to assess the flight performance of classic helicopters. |
| 3 | Helicopter production technology | 1. Knowledge of the technology of helicopter production; <br> 2. Ability to develop technologies for the manufacture of parts and assemblies of helicopters; <br> 3. Knowledge of modern technologies for the development and use of equipment for the production of components, assemblies, and final assembly of helicopters. |

**Table 2.** *Cont*.

| № | Name of the Course | Training Outcomes |
|---|---|---|
| 4 | Operational manufacturability and reliability | 1. Knowledge of the basic provisions of operational manufacturability and reliability;<br>2. Ability to apply domestic and foreign experience to improve the operational manufacturability and reliability of the helicopter and its systems;<br>3. Knowledge of methods and means of ensuring reliability and safety of helicopter operation. |
| **PRP "Project Management"**, program manager: T.I. Zueva | | |
| 5 | Cost management | 1. Knowledge of the classification and components of project costs;<br>2. Knowledge of the cost management methods;<br>3. Ability to plan and analyze the project budget;<br>4. Ability to make decisions regarding cost management. |
| 6 | Project team management | 1. Knowledge of the basics of managing people and small groups;<br>2. Ability to plan and organize team work;<br>3. Ability to resolve conflicts in the team and increase motivation;<br>4. Ability to manage the project team. |
| 7 | Project quality management | 1. Knowledge of international and domestic quality standards;<br>2. Knowledge of the quality management system;<br>3. Ability to organize high-quality execution of work on the project;<br>4. Ability to control the quality of projects and programs at all stages of their life cycle. |
| 8 | Logistic support of projects | 1. Knowledge of the logistic subsystems and the possibilities of their use;<br>2. Ability to organize logistics support of projects;<br>3. Ability to use integrated logistics support for the life cycle of high-technology products. |
| 9 | Project risk management | 1. Knowledge of the main characteristics and types of project risks;<br>2. Ability to apply methods of quantitative and qualitative analysis of project risks and risk management methods;<br>3. Knowledge of risk management skills. |

Let us consider how the content of the PRP is being created on the basis of formulated competencies and the training outcomes. The methodology proposed by the authors is based on the application of the principle of modular construction of the PRP. This implies that for the same PRP, it is possible to use several options of content based on different combinations of modules.

The principles of the disciplines' development are as follows: Each program is supposed to create or improve trainees' specific competencies, relevant knowledge, and skills. That is, as a result of mastering each discipline, a certain competence can be created or improved. In this case, when forming the content of PRP disciplines, it is necessary to take into account the specifics of a particular production.

The competence-based approach to the formation of the content of the PRP allows: Firstly, the development of a program from a set of disciplines that form the competencies necessary for a particular production and meet the requirements of professional standards; secondly, the creation of a sequence of obtaining the competence convenient for both the trainer and the trainee; and, thirdly, the creation of a different set of PRPs for the same groups of trainees, thus giving way to their individual learning trajectories [23]. This approach allow one to pay more attention to students with a low level of training, while reducing the amounts of educational services for more trained ones. The student

can master every discipline of a PRP in accordance with a schedule convenient to them. The learning outcomes for each discipline are recorded, and the student, following the results of their development, acquires the competencies provided for in the PRP. Competencies are finally formed by the students in the preparation and the final certification work defense; then, the student receives a diploma. Thus, this approach allows the organization of an individual learning path for individual categories of students.

It should be noted that, as a rule, when concluding a contract for training with an organization, we are talking about training a group of students. In addition, when recruiting students for PRP training, the following situations are possible:

1.  An employee sent by the customer for training (potential trainee) has already mastered a part of the PRP during the secondary vocational training or higher education, and possesses the appropriate competencies;
2.  One of the employees sent by the customer for training needs to obtain more competencies to fulfill their labor functions than the others.

In this regard, the issue of forming an individual learning trajectory is relevant. An example of the formation of such a trajectory is illustrated in Figure 2.

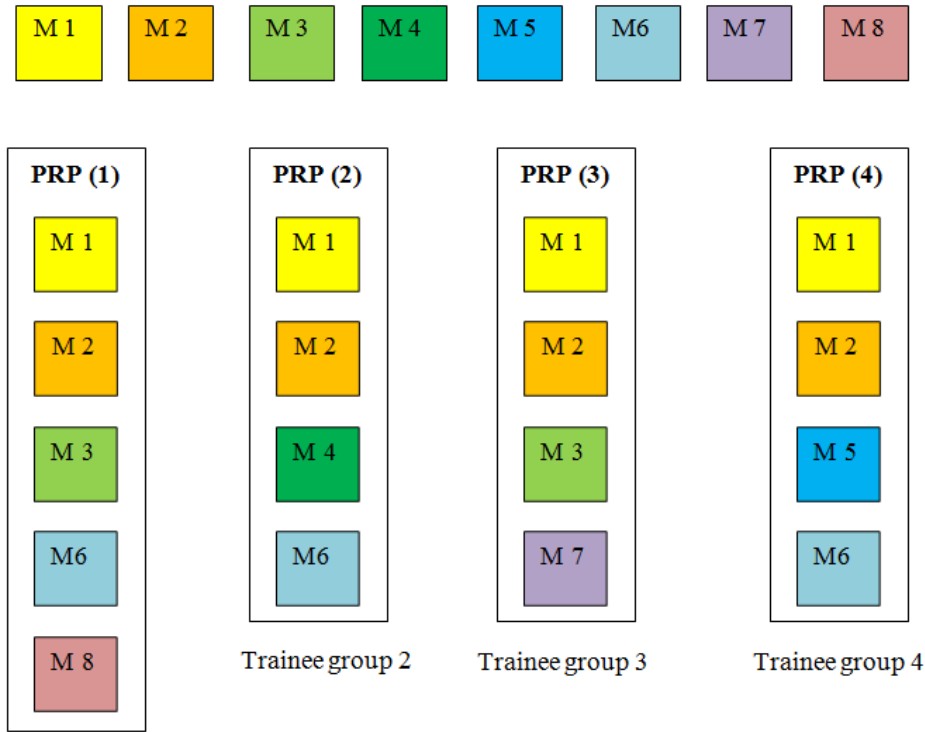

**Figure 2.** Development of the PRP content on a modular basis: M—module.

PRP1, PRP2, PRP3, and PRP4 are four versions of the same PRP, formed according to the modular principle [24,25]. Modules M1 and M2, as we can see in the figure, are basic (common) for all programs. The remaining modules are variable and arranged in the program structure in such a way that it is possible to combine groups of students studying the same module into one group.

Thus, the proposed methodology for developing the contents of a PRP includes the following steps:

1.  Stating the competencies with the customer;
2.  Developing the content of courses to ensure the acquisition of the necessary competence;

3. Developing the content of the PRP by including professional development programs in its structure in accordance with the list of required competencies.

The presented algorithm for developing a PRP allows this process to be formalized. Formal digitalization of the main stages of the PRP development makes the automation of the process possible: According to the requirements of the training customers for the competencies to be formed by students, it is possible to automatically select the necessary disciplines from the existing training base. In this case, automation can be carried out in accordance with the following algorithm:

1. Each advanced training program available in the program database gets one competency (CPRP);
2. A competency base containing the competencies of all programs {$CPRP_1$, $CPRP_2$, ... , $CPRP_m$} is formed;
3. When developing the PRP, the methodologist selects the necessary competencies from the competency base (e.g., $CPRP_1$, $CPRP_2$, $CPRP_3$, $CPRP_4$);
4. For each competency selected by the methodologist, the automated system offers a list of professional development programs targeting the development of specific competencies. However, the same competency may be developed by different programs. In this case, according to the training objectives, the methodologist selects the most suitable professional development program from the proposed list of programs in order to include it in the PRP.

As a result, the number of professional development programs ("courses") included in the structure of the PRP is equal to the number of initial competencies selected by the methodologist and agreed upon with the customer for the PRP. Moreover, additional competencies may be acquired as a result of undergoing several advanced training programs of the PRP.

There may be cases when the competency base is not needed. In this case, an additional discipline should be developed, the content of which should form the necessary competence to fill the gap.

It should be noted that the proposed model of automation of the design process of the programs for professional retraining of employees of enterprises does not exclude program methodologists from the design process, but gives them a tool to facilitate this process.

The proposed technique was implemented at MAI (NRU) using the University's Information and Analytical System. The application of this technique allowed automation of the program development process, thereby reducing development time and simplifying the coordination of planned competencies with the customer in accordance with professional standards.

## 3. Conclusions

The task of developing the content of a PRP (especially given the possibility of individual learning trajectories) appears, as was shown earlier, as a multi-criteria task. When developing a PRP, there is a need to satisfy various—and sometimes quite contradictory—requirements. The algorithm for designing PRPs proposed by the authors allows the simplification of the development process as well as the minimization of the problems that appear during the program design process.

The advantages of the proposed technique are as follows:

1. The PRP design process adapted to the customers' requirements is simplified due to the fact that, based on the competencies to be created that were coordinated with the customer, an automated selection of courses (advanced training programs) from those available in the database is carried out;
2. The transfer of courses for students who have already mastered individual professional development programs that are part of the PRP is facilitated;
3. PRP training is focused on the requirements of professional standards (qualification manuals), which are prerequisites for the human resource services of the customers for employees.

The technique described above was tested at the MAI (NRU) using the University's Information and Analytical System during the process of designing a PRP for industrial workers in aerospace spheres.

**Author Contributions:** Conceptualization, R.R.A. and L.V.B.; methodology, D.A.K.; formal analysis, L.V.B.; data curation, L.V.B. and D.A.K; writing—original draft preparation, R.R.A.; writing—review and editing, R.R.A. All authors have read and agreed to the published version of the manuscript.

**Funding:** This research received no external funding.

**Conflicts of Interest:** The authors declare no conflict of interest.

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
