# Peer review of "Algorithm for Designing Professional Retraining Programs Based on a Competency Approach"

_education, doi:10.3390/educsci10080191_

Round 1

Reviewer 1 Report

The paper presents an interesting and important topic. Besides its importance, the practical employee’s characteristics and skills is indeed crucial to improve the quality of training. The authors suggest applying the competence-based approach used in core curricula training specifically in aerospace industry during the process of the PRP conceptual design. Aerospace industry include not only the collection of manufacturing but also the vehicular flight within and beyond Earth’s atmosphere. As a result, the arrangement of instrument throughout teaching and retaining should be more specific and is not ad hoc. According to this principle, the process of content is made and verified by industrial workers in aerospace spheres.

Overall, the method is convincing and this paper can be accepted.

Author Response

The improvement of existing and the emergence of new technologies for the design and production of aerospace industry facilities necessitates regular retraining of personnel of enterprises and design bureaus. The experience in providing educational services for employees of aerospace industry enterprises has shown that for each enterprise it is necessary to refine the program of such retraining, taking into account the technological features of the production and destination of products. To solve this problem, the authors proposed to create a fairly extensive database of academic disciplines that form specific competencies for students. At the same time, when discussing educational services with the customer, it is not a set of disciplines that are included in it, but the learning outcomes, i.e. the willingness and ability of the employee to perform certain labor functions. And only then, in accordance with the agreed learning outcomes, the content of the program is formed, i.e. the inverse problem is solved.

The authors believe that this approach to the formation of the substantial part of professional retraining programs can be used not only for aerospace industry employees, but also in other industries, the specifics of which can be taken into account when discussing required competencies.

Reviewer 2 Report

Despite being relatively well written, the article is confusing because we cannot understand the scientific methodology of the papper. Clarification is required for publication.

Even in the abstract, I recommend the improvement of it, the clarification of the research methodology and the results.

If the scientific methodology is properly addressed, the article will see its quality improved.

Author Response

A partial explanation of the methodology of the proposed approach to designing professional retraining programs is set forth in the response to the first reviewer.

As for to the remarks: the authors have radically revised the abstract of the article. The proposed methodology allows us to make the process of building competencies more targeted, both within the industry (focused on the needs of a particular enterprise) and within the training group, when the level of initial training of each student can be taken into account and in accordance with it the necessary knowledge and skills that need to train him. It turns out that the proposed methodology allows, if necessary, organizing an individual learning path at any level. At the same time, the result of training is the criterion for a reasonable choice of the disciplines included in the professional retraining programs.

In addition, the formalization of the program design process allows the use of digital technologies, using a database of competencies and disciplines.

The authors are grateful to the reviewers for valuable comments, the result of which was a significant revision of the article, which, in our opinion, improved its quality a lot.

Round 2

Reviewer 2 Report

The review of the paper has undoubtedly resulted in a considerable improvement in the quality of the article.

Just correct minor aspects: 

Line 26: space is missing: "in this paper is" instead of "this paperis"

Line 37: space is missing: "and expectations" nstead of "andexpectations"